# Identification of Potential Hub Genes Related to Aflatoxin B1, Liver Fibrosis and Hepatocellular Carcinoma via Integrated Bioinformatics Analysis

**DOI:** 10.3390/biology12020205

**Published:** 2023-01-29

**Authors:** Hayam Hamdy, Yi Yang, Cheng Cheng, Qizhan Liu

**Affiliations:** 1Center for Global Health, The Key Laboratory of Modern Toxicology, Ministry of Education, School of Public Health, Nanjing Medical University, Nanjing 211166, China; 2Department of Forensic Medicine and Toxicology, Faculty of Veterinary Medicine, New Valley University, New Valley 72713, Egypt

**Keywords:** aflatoxin B1, hepatocellular carcinoma, fibrosis, multi-omics, hub-genes

## Abstract

**Simple Summary:**

Aflatoxin B1 a highly distributed and hepatotoxicant leads to liver damage and the subsequent development of liver cancer. The aim is to study the combined genes and their function in liver damage progression to prevent reversible liver fibrosis which is a dynamic and bidirectional process to become irreversible liver cancer as a result of aflatoxin B1 exposure. The present study revealed that the combined differential expressed genes of AFB1-fibrosis-related and liver cancer-related were connected to cell process disruption, the top ten core genes were identified using four different algorithm methods and the combined core genes showed that the BUB1B and RRM2 genes were core genes of AFB1-liver fibrosis-liver cancer. The inflammatory-related signaling ssGSEA score for BUB1B high expression results in a significant increase in the expression of JAK-STAT regulation and TLR signaling with no effect on the RRM2 gene also the immune checkpoint chemotherapy-related high expression of the BUB1B gene was showed to have a significant change in CTLA4 Blockage in TCHA LIHC patients. Our study will contribute to, the aflatoxin b1 hepatotoxicant damaging progression could be controlled by BUB1B and RRM2 gene study.

**Abstract:**

The molecular mechanism of the hepatotoxicant aflatoxin B1 to induce liver fibrosis and hepatocellular carcinoma (HCC) remains unclear, to offer fresh perspectives on the molecular mechanisms underlying the onset and progression of AFB1-Fibrosis-HCC, which may offer novel targets for the detection and therapy of HCC caused by AFB1. In this study, expression profiles of AFB1, liver fibrosis and liver cancer-related datasets were downloaded from the Gene Expression Omnibus (GEO), and differentially expressed genes (DEGs) were identified by the GEO2R tool. The STRING database, CytoHubba, and Cytoscape software were used to create the protein-protein interaction and hub genes of the combined genes, and the ssGSEA score for inflammatory cells related gene sets, the signaling pathway, and immunotherapy were identified using R software and the GSEA database. The findings revealed that AFB1-associated liver fibrosis and HCC combined genes were linked to cell process disruptions, the BUB1B and RRM2 genes were identified as hub genes, and the BUB1B gene was significantly increased in JAK-STAT signaling gene sets pathways as well as having an immunotherapy-related impact. In conclusion, BUB1B and RRM2 were identified as potential biomarkers for AFB1-induced fibrosis and HCC progression.

## 1. Introduction

Hepatic fibrosis follows persistent liver damage from a variety of causes, such as toxicity, viral infections, autoimmune illnesses, metabolic or genetic disorders, and autoimmune problems. Advanced liver fibrosis patients typically have a bad prognosis since hepatocellular carcinoma (HCC) frequently develops in these patients. HCC is the most common kind of liver cancer, account for 90%. In addition, HCC is the fourth leading cause of cancer-related death globally [1,2]. However, understanding and molecular mechanism on progression from liver fibrosis to HCC have been limited. 

Studies showed that aflatoxin B1 (AFB1) can induce carcinogenesis in the liver, stomach, lung, kidney, rectal colon, breast, and gallbladder [3,4,5,6,7,8], and 4.6–28.2% of hepatocellular carcinoma (HCC) cases are the result of AFB1 exposure [9]. In addition, edidemiological evidence implicates that AFB1 contamination of food in leading cause of human liver cancer. In addition, there is a high prevalence of HCC in the regions with high exposure to AFB1 [10,11]. Although, the diagnosis of liver cancer has advanced significantly in recent years the treatment outcomes remain inadequate [12,13]. To develop therapy options and increase survival percentage of HCC patients, further research is urgently required on the mechanisms underlying HCC development and its micro-environment.

A characteristic of cancer, immune cell infiltration of solid tumors is crucial to tumor development [14]. Immune cells that infiltrate tumors may have their roles altered by the tumor microenvironment to promote tumor growth [15,16]. The pattern of invasive immune cell types is associated with tumor growth and patient prognosis in addition to the characteristics of tumor autonomy [17,18]. For patients with HCC, understanding the changed pattern of immune infiltration during AFB1-Fibrosis-HCC formation and progression is extremely valuable. 

In this study, three microarray datasets of GSE87028, GSE197112, and GSE112790 from the GEO database were used to analyze the differentially expressed genes (DEGs). The TCGA-LIHC cohort was used for immune cell infiltration analysis and survival analysis. GSE87028 is an RNA-seq data set of AFB1-treated HepaRG cells, so it was selected for studying the hepatotoxicity of AFB1. GSE197112 is a data set containing RAN-seq data of patients with liver fibrosis and normal people, and GSE112790 is an RNA-seq data set of HCC patients and normal controls. Through bioinformatics analysis, we aimed to provide new insights into the molecular mechanism underlying AFB1- induced Fibrosis and HCC development and progression, which provide new targets for the diagnosis and treatment of HCC.

## 2. Materials and Methods

### 2.1. Data Source and Identification of DEGs

Three microarray gene expression data sets related to liver-damag (GSE78028, GSE197112 and GSE112790) were selected from the GEO database. In the GSE78028 dataset, human terminally differentiated hepatic HepaRG cells were treated with 0 µM or 25 µM AFB1 and the DEGs were evaluated. In the GSE197112 dataset, gene expression was detected in four patients with liver fibrosis and four normal people to study the altered genes in the progression of human liver fibrosis. In the GSE112790 dataset, a total of 183 liver cancer tumorous tissues versus 15 samples from normal liver were used to study the DEGs. The volcano plots and heatmaps of the DEGs in each dataset were generated using the SRPLOT online tool (http://www.bioinformatics.com.cn/srplot (accessed on 2 December 2022)) with a cutoff value of |log2(FC)| > 2 and *p* < 0.05 considered statistically significant. We got 89 genes by overlapping the DEGs from three GEO datasets.

### 2.2. Gene Ontology (GO) and Kyoto Encyclopedia of Genes and Genomes (KEGG) Analysis

The biological processes of GO enrichment analysis and KEGG pathway enrichment analysis for the 89 combined DEGs among the three GEO datasets above were conducted using the ShinyGO 0.76.2 online tool, which is an intuitive, graphical tool for enrichment analysis. For both GO and KEGG analysis, the top 10 pathways were chosen, and an FDR of less than 0.05 was used as a cut-off criterion. 

### 2.3. Protein-Protein Interaction (PPI) and Hub Gene Discovery

In order to combine biomolecular interaction networks with high throughput expression data and other molecular states into a single conceptual framework, proteins and the functional interaction networks of the 89 combined DEGs were obtained from the STRING database (https://string-db.org (accessed on 17 December 2022)) and graphed in Cytoscape [19]. We utilized the CytoHubba Cytoscape plugin to investigate the hub gene network. This plugin offers resources for investigating key nodes in biological networks by using a variety of methods to recognize hub nodes and the connections between them and other genes.

Different algorithms (MCC, closeness, EPC, and betweenness ranking methods) were chosen in our study to identify the hub genes, and the top 10 hub genes were discovered. The software has created a network with hub nodes colored according to their significance, with red for the highest score and yellow for the lowest score.

The BUB1B and RRM2 genes were found to be the common interacting hub genes between the utilized raking algorithms after the Venn diagram was performed on the top 10 hub genes of each used method. The network analysis was utilized to examine protein-protein interactions with the chosen hub genes, and the SRPLOT online tool was used to display the BP-Go and KEGG enrichment for the interactor network. 

### 2.4. Gene Set Enrichment Analysis (GSEA)

For the purpose of identifying the enriched gene sets based on the highly expressed group of BUB1B and RRM2 genes, the TCGA LIHC samples were split into two groups based on the average value for gene expression. To calculate the enrichment score, 1000 permutations of the KEGG pathways (c2.cp.kegg. v7.1) and the biological process of GO gene sets (c5.bp. v7.1) were utilized (ES). The top 10 highly significant gene sets were chosen after the results were considered statistically significant with |NES| > 1, NOM *p*-value < 0.05, and FDR *q*-value < 0.25. We also employ the Kaplan-Meier technique and Cox regression to assess how BUB1B and RRM2 gene expressions affect prognosis. When performing a Cox regression analysis, variables with *p* < 0.1 in univariate Cox regression were added to multivariate Cox regression, and *p* < 0.05 was regarded as statistically significant. 

### 2.5. Statistical Analysis

R software, version 4.0.5, was used to perform two-sided Spearman correlation analyses between BUB1B and RRM2 genes and immune infiltrating cells in TCGA LIHC samples, which were first analyzed using the xCELL algorithm. For performing the ssGSEA correlation between BUB1B and RRM2 gene expression with memory B cells, immature B cells, effector memory CD4+ T cells, central memory CD4+ T cell immune markers, and T helper 2 cell immune markers gene sets, the “GSVA” package was used. The results of spearman correlation and ssGSEA were visualized using the “tidyverse”, “broom”, “fs” and “lubridate” packages, the area under the ROC curve (AUC) was computed and utilized to compare the diagnostic value of these hub genes using timeROC and survival packages. 

By using GraphPad Prism (version 8.0.1), the difference in BUB1B and RRM2 gene expression between aflatoxin B1 and their control samples, fibrotic and non-fibrotic samples, and liver tumors and normal samples were compared using an independent Student’s *t*-test. Clinically, the difference between high and low BUB1B and RRM2 gene expression and their clinical phenotype data in the TCGA LIHC was analyzed using chi- square test. *p* < 0.05 was considered statistically significant. * *p* < 0.05, ** *p* < 0.01, *** *p* < 0.001.

## 3. Results

### 3.1. Identification and Analysis of DEGs

The microarray results from GEO database were chosen. |log2(FC)| > 2.0 and *p* < 0.05 were used as the standard principal values, and the volcano plots were performed using the SRPLOT tools. The accession number GSE87028, GSE197112 and GSE112790 were selected and the volcano plots for the selected datasets showed that the up-regulated genes were 1846, 1328 and 477 genes, and the down-regulated genes were 2698, 2121 and 453 genes respectively (Figure 1A). In the Venn diagram, the overlap among the 3 selected datasets were 89 DEGs (Figure 1B), and the heatmap for their gene expression on the selected datasets is shown in Figure 1C–E.

The biological categorization of the combined DEGs was examined. The DEGs were significantly enriched in chromatid segregation and microtubule cytoskeleton organization involved in mitosis, according to the results of GO and KEGG pathway analyses (Figure 1F), while the KEGG pathway analysis showed that the DEGs were primarily enriched in DNA replication, base excision, and mismatch repair (Figure 1G). Overall, the combined genes for HCC, liver fibrosis, and AFB1 were connected to cell process disruption. 

### 3.2. Identification and Analysis of Hub Genes

The PPI among the overlapping DEGs was constructed using the STRING online tool (Figure 2A). The query protein was visualized through Cytoscape v3.8.1 as PPI network visualization, and the hub genes were identified by the cytoHubba tool to generate the top-ranked list of proteins. According to the four ranking methods in cytoHubba, the maximal clique centrality (MCC) ranking method, the closeness ranking method, the edge percolated component (EPC) ranking method, and the betweenness ranking method, the hub genes were identified and top 10 hub genes by each method were selected (Figure 2B–E). The intersection was obtained from the genes determined by these four methods, yielding two common interacting Hub genes BUB1 Mitotic Checkpoint Serine/Threonine Kinase B (BUB1B) and Ribonucleotide Reductase Regulatory Subunit M2 (RRM2) (Figure 2F). The mRNA expression of these two genes were increased in AFB1 and liver cancer-related databases, while in liver fibrosis they were decreased (Figure 2G,H). Concluded, the core genes of AFB1-liver fibrosis-HCC were BUB1B and RRM2 genes. so the cell try to decrease the damaging effect and convert it to irreversible damage. 

### 3.3. Functional Enrichment Analysis of Proteins Interacting with BUB1B or RRM2

Then, we identified the functional enrichment analysis of proteins interacting with BUB1B or RRM2 (Figure 3A). Most of these proteins were enriched in the regulation of the mitosis pathway and mitotic cell cycle checkpoint pathway on the gene ontology biological process pathways (GO-BP), and KEGG indicated that most of these proteins are enriched in cell cycle process (Figure 3B,C).The proteins interacting with RRM2 were related to mitosis regulation in GO-BP, and pyrimidine metabolism was the most enriched pathway in KEGG for most of the proteins interacting with RRM2 (Figure 3D–F). 

Following that, we used TCGA LIHC patient samples for additional future verification. For verification of the enrichment results, the GO-BP and KEGG enrichment analyses were identified in the TCGA LIHC patients with BUB1B and RRM2 high expression. It was shown that cytokinetic process, regulation of spindle organization, negative regulation of cell cycle G2/M phase transition, chromosome localization, regulation of ubiquitin protein transferase activity, regulation of chromosome organization, metaphase plate congression, mitotic cell cycle phase transition, regulation of mitotic cell cycle phase transition and cytokinesis were the highly significant enriched GO-BP pathway in BUB1B high expression patients (Appendix A). In RRM2 high expression patients, the GO-BP was enriched in cytoskeleton dependent cytokinesis, cytokinetic process, cytokinesis, regulation of chromosome organization, chromosome localization, mitotic cytokinesis, regulation of mitotic cell cycle phase transition, mitotic cell cycle phase transition, regulation of cell cycle phase transition, DNA replication (Appendix A). 

The KEGG enrichment analysis pathways for BUB1B high expression patient were cell cycle, oocyte meiosis, nucleotide excision repair, DNA replication, mismatch repair, homologous recombination, the p53 signaling pathway, base excision repair, progesterone mediated oocyte maturation, ubiquitin mediated proteolysis (Appendix A). and the RRM2 high expression group shown to have a high enrichment on oocyte meiosis, cell cycle, p53 signaling pathway, DNA replication, homologous recombination, nucleotide excision repair, mismatch repair, progesterone mediated oocyte maturation (Appendix A). In addition, it was showed that the clinicopathological feature between BUB1B and RRM2 genes expression and HCC patients were significantly changed in age, fibrosis Ishak score, neoplasm histologic grade, pathologic T, race and tumor stage diagnoses (Appendix A).

### 3.4. BUB1B and RRM2 Are Related to Immune Cell Infiltration

There are links between BUB1B and RRM2 and immune infiltration, as well as prognostic capacity. To investigate the roles of BUB1B and RRM2 genes in the immune microenvironment of HCC patients, inflammatory cell infiltration was estimated. It was discovered that Th2 cells, pro-B cells, CD4+ memory T cells, and B cells were the highly positively correlated inflammatory cells among the BUB1B and RRM2 genes in TCGA LIHC (Figure 4A). The correlation between BUB1B and RRM2 genes with genes related to selected immune signature gene sets revealed that they had a significantly high positive correlation (Figure 4D–H). Finally, the relationship between inflammatory signaling and immunotherapy gene sets with the expression of BUB1B and RRM2 revealed that the BUB1B high expression group was regarded to have a significantly increased in JAK-STAT regulation gene sets pathway (Figure 5A), while there were no significant differences comparing high and low RRM2 genes (Figure 5C). In addition, the immunotherapy related datasets were significantly increased in high expression of BUB1B group on CTLA4 immunotherapy related gene sets with no effect on PD1 blockade gene sets. While no significant difference in RRM2 gene expression was found between the high and low expression groups (Figure 5B,D). So BUB1B high expression on HCC patients related to changes in the immune-related tumor microenvironment and also to immunotherapy.

### 3.5. Survival Analysis of BUB1B and RRM2 in TCGA-LIHC Cohort

Time-dependent receiver-operating characteristic (ROC) analysis was performed to determine how these markers predict patient survival or death, and it was found that the AUCs (area under the ROC curve) for 1, 2, and 3 years overall survival for the BUB1B gene and the RRM2 gene, respectively, were 0.71, 0.66, and 0.65. The ROC and Kaplan-Meier curves were used to assess their prognostic capacity (Figure 6A,B) Overall survival, disease-specific survival, progression-free interval curve, and disease-free interval curve than patients in the low-risk group (Figure 6C,D).

For TCGA LIHC patients, we conducted univariate and multivariate Cox regression survival analyses to ascertain if the risk score plays an independent prognostic role. An increased risk score was strongly connected with worse pathogenic T, N, and M in LIHC patients, according to the results of the univariate Cox regression analysis. After that, multivariate analysis showed that the BUB1B and RRM2 genes had a significantly independent predictive value for overall survival, disease-specific survival, and progression-free interval (Figure 6E–H) (Appendix A). After controlling for confounding factors, these findings revealed that the BUB1B and RRM2 genes were a separate predictive factor for patients with LIHC. 

## 4. Discussion

In recent years, despite great progress in the identification and analysis of AFB1 and its damaging effects on the liver, we cannot prevent or decrease its irreversible hazard effect on the liver to be converted to HCC, and its mortality remains unacceptably high. Moreover, studies that focus on understanding their function in HCC are less clear. We hoped to gain new insights into the molecular mechanisms underlying AFB1-induced fibrosis and HCC development and progression through bioinformatics analysis, which should lead to new targets for HCC diagnosis and treatment. The result showed that the BUB1B and RRM2 genes were hub genes in AFB1-liver fibrosis-HCC progression.

Our results indicate that BUB1B was significantly decreased in liver fibrosis and increased in AFB1 and liver cancer samples, for the reason that fibrosis is reversible liver damage and the body tries to protect itself from being damaged. It was stated that a decreasing APC/C inhibitory protein like BUB1B triggers the activation of APC/C at mitosis and it plays an essential function during cell proliferation by preventing the re-replication of DNA [20,21].

The increased expression of BUB1B was considered to facilitate an inaccurate DNA repair process termed “alternative non-homologous end joining that inaccurately repairs DNA damages [22]. The RRM2 protein interact with key cell cycle genes and signaling pathway proteins, such as p53, PI3K, hypoxia inducible factor-1α, Bax, and cyclin D1, to regulate tumor cell proliferation, migratory, and invasive abilities and cell cycle progression [23,24,25,26]. Moreover, the reduction or inhibition in human cancer cells resulted in massive chromosome loss and apoptotic cell death [27]. In addition, as tumors are “fibrotic wounds that do not heal” and chronic fibrosis is a risk factor for cancer [28,29]. Based on these findings, the hepatic cell is damaged and fibrosis occurs when exposed to aflatoxin B1 and has an increase in Bub1b expression. However, the cell tries to protect itself as a defense mechanism, which results in a decrease in BUB1B and RRM2 expressions. However, when there is extensive damage caused by AFB1, the cell cannot compensate for all of these damages, and an increase in gene expression and inaccurate DNA repairing occurs, resulting in liver cancer. So, while we can control the progression of the liver-damaging process by monitoring BUB1B and RRM2 gene expression and preventing the progression of reversible liver fibrosis to irreversible liver cancer as a result of AFB1 exposure, we still need to understand the functional roles and potential mechanisms of BUB1B and RRM2 in liver fibrosis and liver cancer progression as a result of AFB1 exposure.

It is important to define the precise roles and known molecular mechanisms of BUB1B in the initiation and progression of HCC. BUB1B was shown to be elevated in HCC tissues as well as HCC cell lines, BUB1B overexpression and unfavorable clinicopathological characteristics were positively correlated; in addition, lower recurrence-free and overall survival rates were linked to BUB1B overexpression in HCC patients, per survival studies [30]. The functional study revealed that the up-regulated BUB1B helped HCC cells proliferate, migrate, invade, and metastasize [31]. It has also been found to be a crucial part of the spindle assembly checkpoint in mitosis, which delays the start of anaphase and ensures normal chromosomal segregation [32]. According to a recent study, JNK/c-Jun signaling activity was crucial for carcinogenicity [33].

According to the present study, the majority of immune cells in HCC were significantly positively correlated with BUB1B and RRM2. These indicators were also strongly positively related to immune cell invasion. These findings imply that tumor immune infiltration may be crucial to the RRM2-mediated progression of HCC. Immune checkpoint blockade medications work best when immune checkpoint molecules are expressed in tumor tissues [34,35,36].

Immune checkpoint receptors have the ability to either suppress or stimulate immune response mechanisms [37]. The relationship between RRM2 and immunological checkpoints was therefore examined. Moreover, a significant relationship between CTLA-4 and BUB1B expression in HCC, suggesting that tumor immune evasion may influence the hepato-carcinogens that BUB1B mediates. It is stated that in the TME, infiltrating immune cells form an ecosystem and they play an important role in tumor progression and have important prognostic value [38]. In addition, Studies have shown that cytotoxic CD8+ T cells and CD4+ helper T cells can target antigenic tumor cells and inhibit tumor cell growth [39]. In summary, the combined differential expressed genes of AFB1- fibrosis related and liver cancer related were connected to cell process disruption. the top ten core genes were identified using four different algorithm methods and the combined the combined core genes showed that the BUB1B and RRM2 genes were core genes of AFB1-liver fibrosis-HCC. Moreover, the expression of those genes was up regulated in AFB1 and liver cancer related while they were down regulated in fibrosis related. The inflammatory cell found to have a correlation with both BUB1B and RRM2 genes in TCGA LIHC patients. The inflammatory related signaling ssGSEA score for BUB1B high expression have a significantly increasing in the expression of JAK-STAT regulation and TLR signaling with no effect on RRM2 gene also the immuno-check point chemotherapy related for high expression of BUB1B gene showed to have a significant change in CTLA4 Blockage in TCHA LIHC patients.

Our study has limitations, and a bigger cohort is needed to further validate these findings. Additionally, more thorough research is required to confirm the roles of the BUB1B and RRM2 genes in the development of AFB1-fibrosis and HCC as well as their effects on immunity using in vitro and in vivo assays. This will improve the precision of diagnosis and prognosis and may help with the creation of a targeted therapy for LIHC.

## Figures and Tables

**Figure 1 biology-12-00205-f001:**
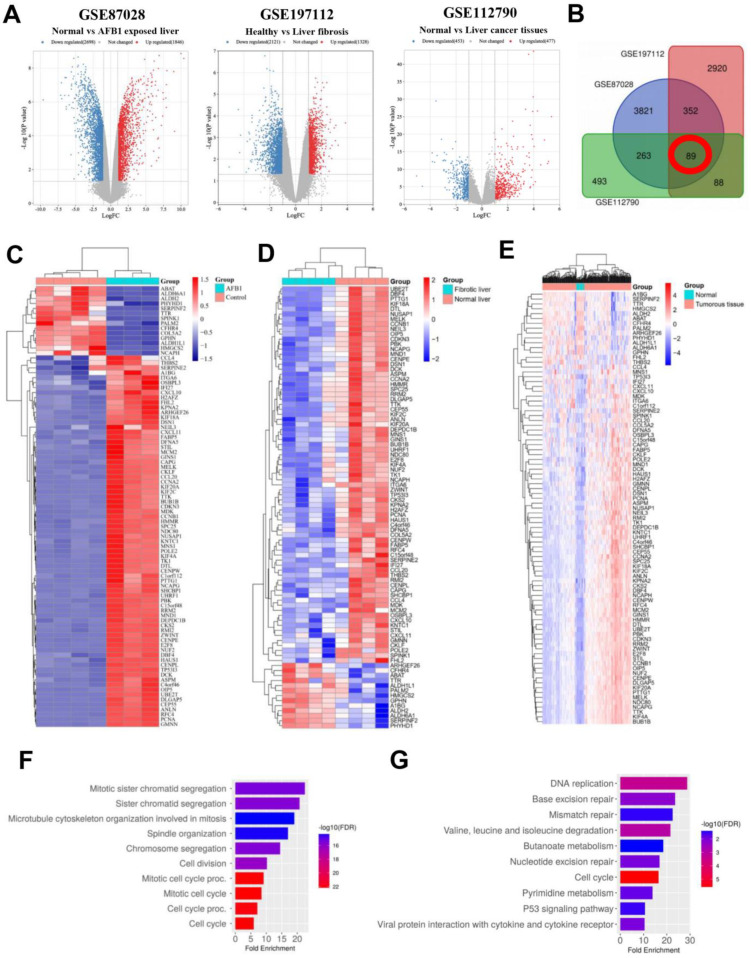
Differentially expressed genes (DEGs) and functional enrichment analysis in AFB1-exposed, liver fibrosis and liver cancer datasets. (**A**) Volcano plot of DEGs in GSE87028, GSE197112 and GSE112790 datasets. Red color indicated up-regulated genes and blue color indicated down-regulated genes. (**B**) Venn diagram of overlapping DEGs among three GEO datasets. (**C**–**E**) Heat maps of overlapping DEGs in GSE87028, GSE197112 and GSE112790 datasets. (**F**) Gene ontology (GO) biological processes (BP) enrichment analysis of 89 overlapping DEGs. (**G**) Kyoto encyclopedia of genes and genomes (KEGG) pathway analysis of 89 overlapping DEGs.

**Figure 2 biology-12-00205-f002:**
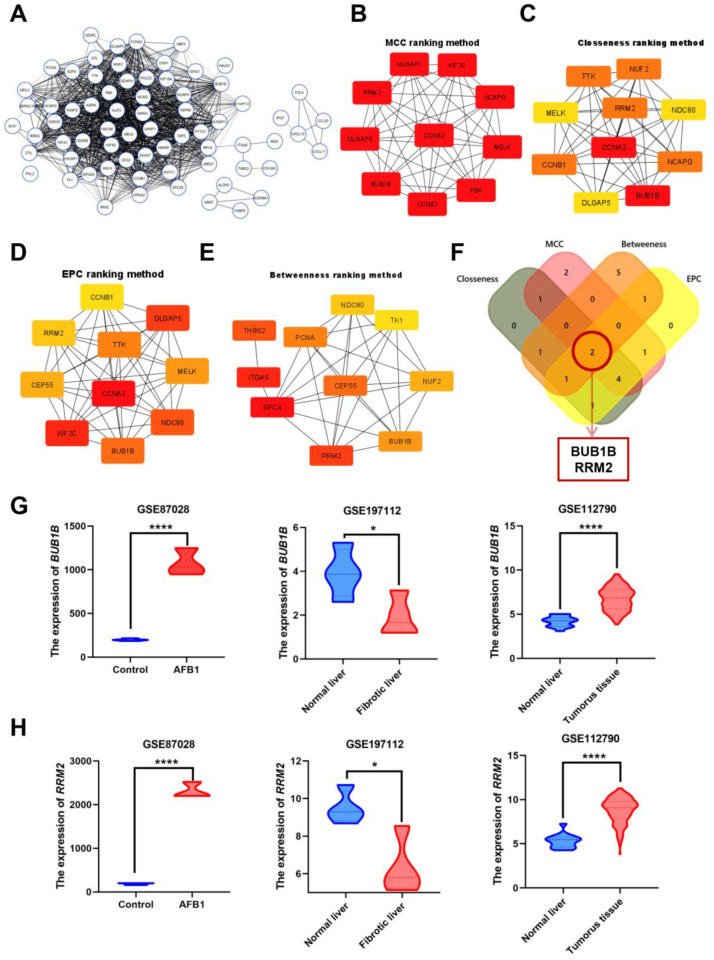
Identification of Hub genes in overlapping DEGs among three GEO datasets. (**A**) Protein-protein interaction of the overlapping DEGs. (**B**–**E**) Maximal clique centrality (MCC) ranking method, closeness ranking method, edge percolated component (EPC) ranking method, and betweenness ranking method for hub genes identification. (**F**) Venn diagram for identifying hub genes among different ranking methods. (**G**) The expression of BUB1B in three GEO datasets. (**H**) The expression of RRM2 in three GEO datasets. Red color represents highest degree, and orange color represents intermedia degree, and yellow color represents lowest degree. * *p* < 0.05; **** *p* < 0.0001.

**Figure 3 biology-12-00205-f003:**
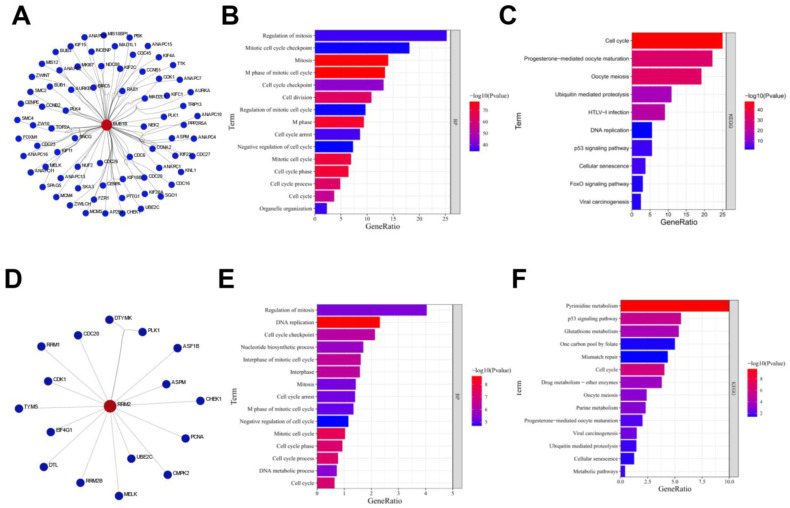
Functional enrichment analysis of proteins interacting with BUB1B or RRM2. (**A**) Protein interaction network of BUB1B. (**B**) BP enrichment analysis of protein interaction network of BUB1B. (**C**) KEGG pathway analysis of protein interaction network of BUB1B. (**D**) Protein interaction network of RRM2. (**E**) BP enrichment analysis of protein interaction network of RRM2. (**F**) KEGG pathway analysis of protein interaction network of RRM2.

**Figure 4 biology-12-00205-f004:**
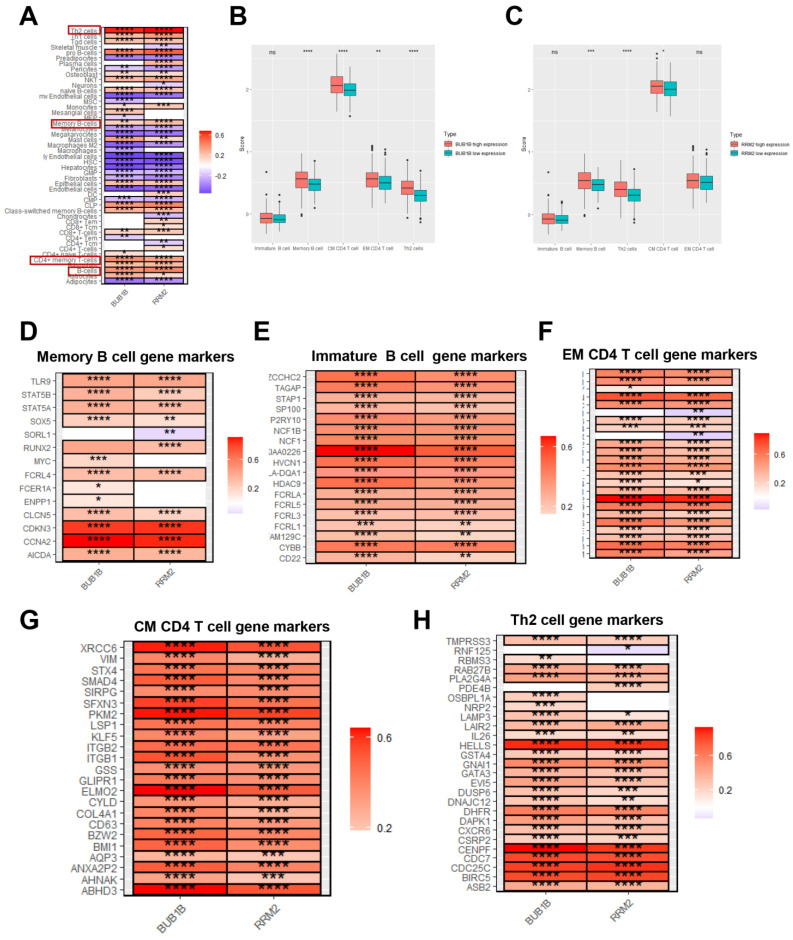
BUB1B and RRM2 are related to immune cell infiltration. (**A**) The correlation between immune cell infiltration and the expression of BUB1B and RRM2 in TCGA-LIHC cohort. (**B**) Comparison of ssGSEA scores between BUB1B high expression group and BUB1B low expression group in TCGA-LIHC cohort. (**C**) Comparison of ssGSEA scores between RRM2 high expression group and RRM2 low expression group in TCGA-LIHC cohort. (**D**–**H**) The correlation between memory B cells immune markers, immature B cells immune markers, effector memory CD4+ T cell immune markers, central memory CD4+ T cell immune markers, and Th2 cell immune markers with the expression of BUB1B and RRM2. * *p* < 0.05; ** *p* < 0.01; *** *p* < 0.001; **** *p* < 0.0001.

**Figure 5 biology-12-00205-f005:**
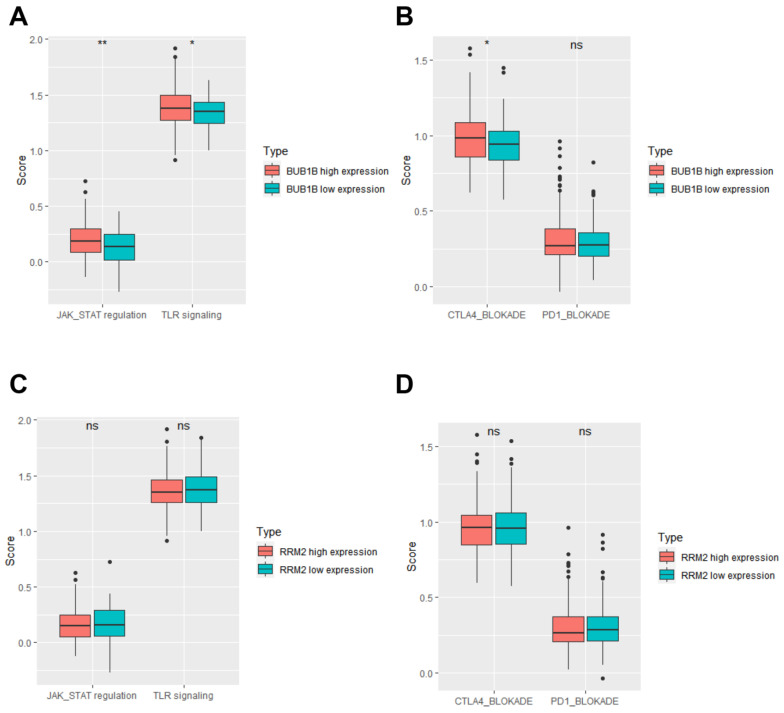
Analysis of the relationship between inflammatory signaling and cancer therapy gene sets with the expression of BUB1B and RRM2 in TCGA-LIHC cohort. (**A**,**B**) Comparison of JAK-STAT regulation gene sets, TLR signaling gene sets, and CTLA4 and PD1 blockade cancer immune therapy gene sets between BUB1B high expression group and BUB1B low expression group. (**C**,**D**) Comparison of JAK-STAT regulation gene sets, TLR signaling gene sets, and CTLA4 and PD1 blockade cancer immune therapy gene sets between RRM2 high expression group and RRM2 low expression group. * *p* < 0.05; ** *p* < 0.01.

**Figure 6 biology-12-00205-f006:**
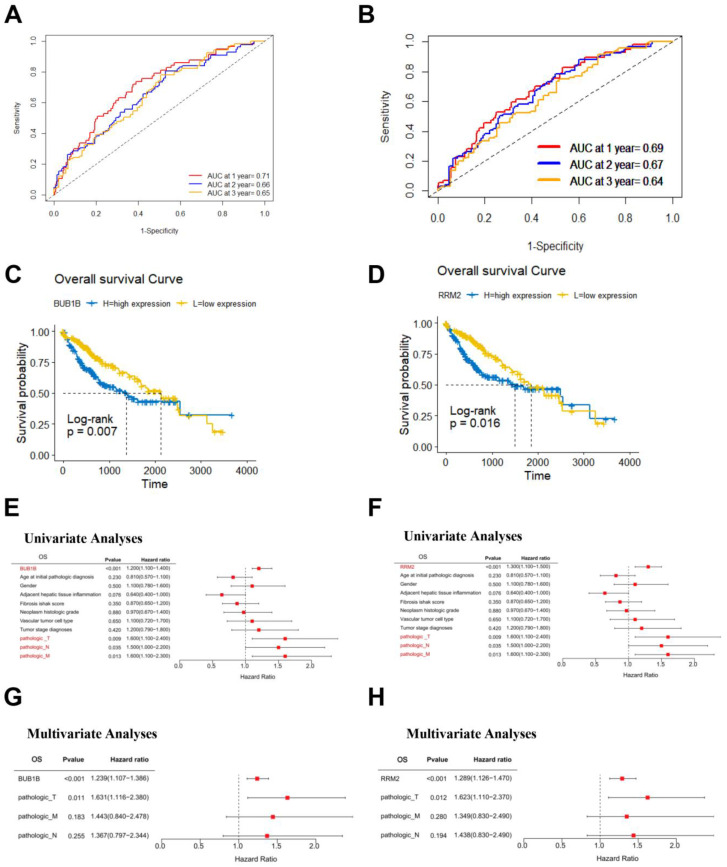
Survival analysis of BUB1B and RRM2 in TCGA-LIHC cohort. (**A**,**B**) ROC curves of TCGA-LIHC cohort. The AUC values shown the predictive efficiency of BUB1B and RRM2 on the 1-, 2-, and 3-years survival rate. (**C**,**D**) The overall survival analysis of BUB1B and RRM2 in TCGA-LIHC cohort. (**E**–**H**) Univariate and multivariate Cox analyses evaluated the independent prognostic value of BUB1B and RRM2 in terms of OS in TCGA-LIH.

## Data Availability

The data used to support our results are available at the GEO (https://www.ncbi.nlm.nih.gov/geo/ accessed on 1 July 2022), TCGA (https://portal.gdc.cancer.gov/ accessed on 15 July 2022).

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
