# Peer review of "Identification of Potential Hub Genes Related to Aflatoxin B1, Liver Fibrosis and Hepatocellular Carcinoma via Integrated Bioinformatics Analysis"

_biology, 2023, doi:10.3390/biology12020205_

Round 1

Reviewer 1 Report

I would suggest for the abstract section to avoid using enumeration (1.Background, 2.Purpose, 3.Methods, etc).

In my opinion the references list is too short, I suggest to add few more titles.

The Conclusion section does not exist in the manuscript, however from line 314 to 321 could be considered the final conclusion of this paper, specially that some of the information could be found in the conclusion section of the Abstract.

Author Response

Point 1:  I would suggest for the abstract section to avoid using enumeration (1.Background, 2.Purpose, 3.Methods, etc).

Response1: Thank you! We found your suggestion extremely helpful and have changed it accordingly.

Point 2: In my opinion the references list is too short, I suggest to add few more titles

Response 2: We agree that better to add much more references, would be taken your kindly opinion.

Changes: We add additional references in the introduction and discussion parts. The adding information and their references appear in red type by track changes in the revised paper.

Point 3: The Conclusion section does not exist in the manuscript, however from line 314 to 321 could be considered the final conclusion of this paper, specially that some of the information could be found in the conclusion section of the Abstract.

Response 3: Thank you for this excellent observation.

Changes: it is added in a red color in the revised paper.

Reviewer 2 Report

The authors used bioinformatics analysis to identify Hub genes involved in Aflatoxin B1, cirrhosis, and hepatocellular carcinoma.

As a result, two genes, BUB1B and RRM2, were identified.

The experiments are well done, with detailed analysis, and the supplement tables are well presented.

I would like to see a careful discussion of the following points in the Discussion.

Why are the values of AFB1 and tumorous tissues higher than normal and fibrotic tissues lower than normal? And this is not AFB1 -> fibrosis -> HCC (isn't fibrosis -> HCC only with hepatitis virus?) and not AFB1 -> HCC (no toxicity to fibrosis, only carcinogenic effect? This is an area that needs to be considered.

As for the results related to the immune system or immune checkpoint inhibitors shown in Figure 5, there are differences in the two genes. More detailed discussion of this issue, including the role of the genes, will be necessary.

Author Response

Point 1: Why are the values of AFB1 and tumorous tissues higher than normal and fibrotic tissues lower than normal? And this is not AFB1 -> fibrosis -> HCC (isn't fibrosis -> HCC only with hepatitis virus?) and not AFB1 -> HCC (no toxicity to fibrosis, only carcinogenic effect? This is an area that needs to be considered

Response 1: We thank the reviewer for pointing out this important issue. The increased expression of BUB1B was considered to facilitate an inaccurate DNA repair process termed "alternative non-homologous end joining that inaccurately repairs DNA damages (Komura et al., 2021). The RRM2 protein interact with key cell cycle genes and signaling pathway proteins, such as p53, PI3K, hypoxia inducible factor-1α, Bax, and cyclin D1, to regulate tumor cell proliferation, migratory, and invasive abilities and cell cycle progression (Li et al., 2018; Shah et al., 2015; Rahman et al., 2013 and Wang et al., 2018). Moreover,the reduction or inhibition in human cancer cells resulted in massive chromosome loss and apoptotic cell death (Kops et al., 2004). In addition, as tumors are "fibrotic wounds that do not heal" and chronic fibrosis is a risk factor for cancer (Dvorak, 2015 and  Rybinski, et al., 2014). Based on these findings, the hepatic cell is damaged and fibrosis occurs when exposed to aflatoxin B1 and has an increase in Bub1b expression. However, the cell tries to protect itself as a defense mechanism, which results in a decrease in BUB1B and RRM2 expressions. However, when there is extensive damage caused by AFB1, the cell cannot compensate for all of these damages, and an increase in gene expression and inaccurate DNA repairing occurs, resulting in liver cancer. So, while we can control the progression of the liver-damaging process by monitoring BUB1B and RRM2 gene expression and preventing the progression of reversible liver fibrosis to irreversible liver cancer as a result of AFB1 exposure, we still need to understand the functional roles and potential mechanisms of BUB1B and RRM2 in liver fibrosis and liver cancer progression as a result of AFB1 exposure.

Dvorak, H. F.; Tumors: wounds that do not heal-redux, Cancer Immunol. Res 2015, 3 (1)  1–11.

Komura K.; Inamoto T.; Tsujino T.;  Matsui Y.;  Konuma T.; Nishimura K, et al.. "Increased BUB1B/BUBR1 expression contributes to aberrant DNA repair activity leading to resistance to DNA-damaging agents". Oncogene 2021; 40 (43): 6210–6222. doi:10.1038/s41388-021-02021-y.

Kops, G. J.; Foltz, D. R. & Cleveland, D. W. Lethality to human cancer cells through massive chromosome loss by inhibition of the mitotic checkpoint. Proc. Natl Acad. Sci. USA 2004, 101, 8699–8704 .

Li , C.; Zheng, J.; Chen, S.; Huang, B.; Li, G.; Feng, Z.;  Wang, J.;  Xu, S. RRM2 promotes the progression of human glioblastoma. J Cell Physiol. 2018; 233:6759–67. https://doi.org/10.1002/jcp.26529.

Rahman, M. A.;  Amin, A. R.;  Wang, D.;  Koenig, L.;  Nannapaneni,  S,  Chen,  Z.;  Wang,  Z.; Sica, G.;  Deng, X.;  Chen,  Z. G.;  Shin,  D. M. RRM2 regulates bcl-2 in head and neck and lung cancers: a potential target for cancer therapy. Clin Cancer Res. 2013; 19:3416–28. https://doi.org/10.1158/1078-0432.CCR-13-0073.

Rybinski, B.; Franco-Barraza,  J.;  Cukierman, E.;  The wound healing, chronic fibrosis, and cancer progression triad, Physiol. Genomics 2014 , 46 (7)  223–244.

Shah, K. N.; Wilson, EA.; Malla, R.; Elford,  H. L.;  Faridi, J. S. Targeting ribonucleotide reductase M2 and NF-κB activation with didox to circumvent tamoxifen resistance in breast cancer. Mol Cancer Ther. 2015; 14:2411–21. https://doi.org/10.1158/1535-7163.MCT-14-0689.

Wang,  N.; Li, Y.; Zhou, J. Downregulation of ribonucleotide reductase subunits M2 induces apoptosis and G1 arrest of cervical cancer cells. Oncol Lett. 2018; 15:3719–25. https://doi.org/10.3892/ol.2018.7806

Point 2: As for the results related to the immune system or immune checkpoint inhibitors shown in Figure 5, there are differences in the two genes. More detailed discussion of this issue, including the role of the genes, will be necessary.

Response 2: We appreciate the consideration of the  related to the immune system or immune checkpoint inhibitors.

Changes:  we add a section on the discussion part related to the immuno related check point; it was found that a significant relationship between CTLA-4 and BUB1B expression in HCC, suggesting that tumor immune evasion may influence the hepato-carcinogens that BUB1B mediates. It is stated that in the tumor micro environment, infiltrating immune cells form an ecosystem and they play an important role in tumor progression and have important prognostic value. (Petitprez et al., 2020). In addition, Studies have shown that cytotoxic CD8+ T cells and CD4+ helper T cells can target antigenic tumor cells and inhibit tumor cell growth (Sharma and Allison 2015)

Petitprez, F.; Meylan, M.; De Reyniès, A.;  Sautès-Fridman, C.; Fridman, W. H. The tumor microenvironment in the response to immune checkpoint blockade therapies. Front Immunol. 2020, 11:784. 10.3389/fimmu.2020.00784.

Sharma, P.; Allison, J.P. The future of immune checkpoint therapy. Science. 2015,  348:56–61.

Reviewer 3 Report

The manuscript of Hayam Hamdy et al. is of interest, and I commend the Authors for their nice work. My comments are:

1.     It would be interesting to have a summary section.

2.     The section of ethical statement is highly recommended.

3.     Please, improve the quality of the figures.

4.     Please, re-check the punctuation, syntax, and grammar throughout the manuscript.

5.     Please check all references according to the journal instructions.

Author Response

Point 1: It would be interesting to have a summary section.

Response 1: Thank you for reminding us how important it is and we add that part in the revised manuscript.  In summary, the combined differential expressed genes of AFB1- fibrosis related and liver cancer related were connected to cell process disruption.  the top ten core genes were identified using four different algorithm methods and the combined the combined core genes showed that the BUB1B and RRM2 genes were core genes of AFB1-liver fibrosis-HCC. Moreover, the expression of those genes were up regulated in AFB1 and liver cancer related while they were down regulated in fibrosis related. The inflammatory cell found to have a correlation with both BUB1B and RRM2 genes in TCGA LIHC patients. The inflammatory related signaling ssGSEA score for BUB1B high expression have a significantly increasing in the expression of JAK-STAT regulation and TLR signaling with no effect on RRM2 gene also the immuno-check point chemotherapy related for high expression of BUB1B gene showed to have a significant changes in CTLA4 Blockage in TCHA LIHC patients.

Point 2: The section of ethical statement is highly recommended.

Response 2: Thank you for reminding, we add this section in the revised manuscript.

Point 3: Please, improve the quality of the figures.

Response 3: We have revised the figures to provide higher resolution so that their contents can be clearly identified.

Point 4: Please, re-check the punctuation, syntax, and grammar throughout the manuscript.

Response 4: we re-check the punctuation, syntax, and grammar throughout the manuscript.

Point 5: Please check all references according to the journal instructions.

Response 5: We  check the old and new references .